# Tic-Tac: A Translational Approach in Mechanisms Associated with Irregular Heartbeat and Sinus Rhythm Restoration in Atrial Fibrillation Patients

**DOI:** 10.3390/ijms241612859

**Published:** 2023-08-16

**Authors:** Alfredo Parra-Lucares, Eduardo Villa, Esteban Romero-Hernández, Gabriel Méndez-Valdés, Catalina Retamal, Geovana Vizcarra, Ignacio Henríquez, Esteban A. J. Maldonado-Morales, Juan H. Grant-Palza, Sofía Ruíz-Tagle, Victoria Estrada-Bobadilla, Luis Toro

**Affiliations:** 1Critical Care Unit, Department of Medicine, Hospital Clínico Universidad de Chile, Santiago 8380420, Chile; 2Cardiovascular Department, Hospital Clínico Universidad de Chile, Santiago 8380420, Chile; 3School of Medicine, Faculty of Medicine, Universidad de Chile, Santiago 8380420, Chile; 4MD PhD Program, Faculty of Medicine, Universidad de Chile, Santiago 8380420, Chile; 5Division of Internal Medicine, Department of Medicine, Hospital Clínico Universidad de Chile, Santiago 8380420, Chile; 6Division of Nephrology, Department of Medicine, Hospital Clínico Universidad de Chile, Santiago 8380420, Chile; 7Centro de Investigación Clínica Avanzada, Hospital Clínico, Universidad de Chile, Santiago 8380420, Chile

**Keywords:** atrial fibrillation, electrophysiology, ablation therapy

## Abstract

Atrial fibrillation (AF) is a prevalent cardiac condition predominantly affecting older adults, characterized by irregular heartbeat rhythm. The condition often leads to significant disability and increased mortality rates. Traditionally, two therapeutic strategies have been employed for its treatment: heart rate control and rhythm control. Recent clinical studies have emphasized the critical role of early restoration of sinus rhythm in improving patient outcomes. The persistence of the irregular rhythm allows for the progression and structural remodeling of the atria, eventually leading to irreversible stages, as observed clinically when AF becomes permanent. Cardioversion to sinus rhythm alters this progression pattern through mechanisms that are still being studied. In this review, we provide an in-depth analysis of the pathophysiological mechanisms responsible for maintaining AF and how they are modified during sinus rhythm restoration using existing therapeutic strategies at different stages of clinical investigation. Moreover, we explore potential future therapeutic approaches, including the promising prospect of gene therapy.

## 1. Introduction

Atrial fibrillation (AF) is the most common sustained cardiac arrhythmia worldwide, with a global prevalence of around 1%, reaching nearly 10% in individuals over 80 years old [1]. It is characterized by a chaotic atrial rhythm derived from irregular electrical activity in different ectopic trigger sites, such as the pulmonary veins (PV) and the left atrium (LA) [2]. It is a significant cause of disability and death, especially in the elderly population, due to the development of complications, such as strokes, frequent hospitalizations, and bleeding associated with anticoagulant therapy [3]. This population is highly relevant, given that the primary risk factor for the onset of AF is age [4].

The therapeutic approach in AF has been a subject of debate for decades regarding the management of heart rate versus heart rhythm [5]. Initially, it was established that there was no difference between one therapeutic approach and another, leading to a preference for rate control over rhythm control due to the low rate of effectiveness and the proarrhythmic complications associated with antiarrhythmic drugs (AAD) [6]. However, over time, evidence began to demonstrate that AF progresses through different stages of the disease via electrical and structural remodeling phenomena that intensify with the passage of years [7]. Recent studies have highlighted the advantages of early conversion to sinus rhythm in patients with this arrhythmia [8,9,10].

Based on the above, it can be concluded that the restoration of sinus rhythm may reverse a series of electrophysiological and molecular mechanisms that underlie the natural progression of the disease, through which a patient transitions from paroxysmal AF episodes to the final stage of permanent AF [11].

The electrophysiological mechanisms involve alterations in the composition of ion channels, leading to the generation of a cellular substrate that facilitates the onset of arrhythmia [12]. This promotes the formation of well-characterized trigger sites [2] that initiate the episodes, while their maintenance is attributed to existing micro re-entry circuits [13]. Similarly, the autonomic nervous system modulates the electrical activity and induces changes in target proteins involved in membrane potential, calcium release and uptake cycles, or calcium transients [14]. The perpetuation of these phenomena leads to the emergence of additional trigger sites and chaotic atrial contractions during the advanced stages unless timely intervention is provided to restore normal electrical conduction [7].

Currently, additional mechanisms have emerged to complement the arrhythmogenic substrate, shedding light on the perpetuation of arrhythmia. These include factors, such as inflammation [15], fibrosis [16], altered gap junctions [17], and genetic predisposition [18,19], which are yet to be fully characterized across the different stages of the disease. Furthermore, there has been a re-assessment of the classical multiple wavelet hypothesis, demonstrating that the substrates of the disease at the clinical level are focal trigger sites of action potentials rather than multiple micro re-entry sites [20], which provides insights into the mechanisms involved in electrophysiological ablation-based therapy.

The objective of this article is to provide a clinical-molecular characterization of AF and its mechanisms, and how they are modified during sinus rhythm restoration with therapeutic strategies involving pharmacological approaches and electrophysiological ablation. Additionally, we aim to provide a future perspective based on gene therapy in AF, emphasizing the importance of early reversal from irregular and chaotic rhythm to sinus rhythm in our patients.

## 2. The Hallmarks of Atrial Fibrillation

The clinical classification of AF is principally based on whether the episode terminates spontaneously or not, and the duration of the episode, progressing from paroxysmal, persistent, long persistent to permanent AF [1]. Recently, new classification systems for AF have been proposed [21]. Nonetheless, their pathophysiological correlate is still uncertain. The classical clinical categories have proven distinct behaviors that may reflect the underlying molecular mechanisms [22], although they are still not fully understood.

In the following section, we describe the pathophysiological continuum that explains the origin, maintenance, progression, and stabilization of AF, with the key aspects that have been described for each of the clinical stages, which will be referred to as “hallmarks”. We also propose an “at risk” stage, analogous to what exists for heart failure (HF) [23] that is based on the genetic susceptibility and non-AF remodeling that may lead in some people to the development of this disease. It must be noted that AF has several clinical presentations that do not always follow this order (Figure 1).

### 2.1. At Risk for Atrial Fibrillation

#### 2.1.1. Genetic Substrate

To develop AF, the well-known ectopic activity needs a histological and electrophysiological substrate. Several genetic polymorphisms and mutations have been identified that encode genes associated with ion channel function, calcium handling, transcription factors, and cardiac development and function that have been described elsewhere [12]. Nonetheless, for most people, these associations do not explain the entirety of the disease and only place them as a group that is at risk of developing AF [19]. Familial AF adds up to only 10–15% of AF cases [12], and generally present an earlier onset, in people that may lack traditional risk factors, which has also been called lone-AF. These monogenic paradigms have helped to gain understanding of the mechanisms that generate AF in an otherwise healthy tissue.

#### 2.1.2. Age-Related Remodeling

Age is one of the most well-known risk factors for developing AF, with a lifetime risk of 40% and 50% of patients with AF aged 75 and older [24]. Age-related cardiomyocyte loss produces fibrosis [25], and the automaticity in the sinoatrial node involves hyperpolarization-activated cyclic nucleotide-gated (HCN) channels, whose expression ha shown an age-dependent increase in animal models, which may have a role in atrial ectopy [11]. Other electrophysiological and conduction changes that lead to Ca^2+^ mishandling, prolongation in the action potential duration (APD), effective refractory period (ERP) depolarization, and connexin downregulation also promote arrhythmogenesis [26]. In addition, older patients have a higher incidence of cardiovascular comorbidities like hypertension, HF, and valvular heart disease, and their atrial tissue has been exposed for a longer period to different stressors and remodeling processes [25].

#### 2.1.3. Disease-Related Remodeling

Environmental factors play a key role in the development of AF in most patients. It is estimated that hypertension, in particular high systolic blood pressure, is responsible for 14% of all cases of AF [27]. Obesity provides a proinflammatory, metabolic, and comorbid context that leads these patients to be at high risk of developing cardiovascular disease (CVD) [28]. Insulin resistance is linked to endothelial dysfunction that results from a mismatch between mitogen-activated protein kinase (MAPK) and phosphatidylinositol 3-kinase (PI3K) pathways, which increases endothelial cell death and inflammation [18], and in diabetes, hyperglycaemia leads to advanced glycation end products that lead to fibrosis and hypertrophy. Other factors like lifestyle and diet have shown harmful, but also potentially protective effects in susceptibility to AF [18]. Regarding cardiovascular comorbidities, HF is associated with increased atrial filling pressure that lead to atrial dilation and wall stretch. This histologically translates to a “tear and scar” mechanism that generates fibrosis [29]. A similar situation occurs with valvular heart diseases.

Current classification includes subclinical AF defined as an individual without symptoms attributable to AF, in whom AF is not previously diagnosed and presents an atrial high-rate episode detected by an insertable cardiac or wearable monitor [1]. It is still not clear in which patients the use of devices for primary prevention might be beneficial, but many predictors and risk factors have been identified that include but are not limited to the aforementioned [30]. Guidelines recommend an opportunistic screening in patients with 65 years old or more [1]. Nonetheless, considering the rapidly growing wearable technology and artificial intelligence AF models, we believe that defining a group of patients at risk of AF, which has a clear pathophysiological basis, may prove useful for early detection strategies of AF.

### 2.2. Paroxysmal Atrial Fibrillation

During normal atrial action potential, the cardiac voltage-dependent Na^+^ channel produces the depolarizing current (I_Na_) that triggers the activation of L-type Ca^2+^ channels that is responsible for the calcium-induced calcium release from the sarcoplasmic reticulum (SR). Ca^2+^ is released by ryanodine receptors (RyR2), its uptake into the SR is mediated by the SR Ca^2+^-adenosine triphosphate (SERCA2a), and inside the SR it is bound to calsequestrin. SERCA2a function is limited by the inhibitory subunit phospholamban. Meanwhile, delayed-rectifier K^+^ currents (I_Kr_, I_Ks_, I_Kur_) and the transient-outward K^+^ current (I_to_) control the repolarization and determine the action potential duration (APD) [31].

The hallmarks in paroxysmal AF are the focal ectopic activity and a re-entry substrate prone to abnormal conduction, that are major determinants in AF onset and perpetuation [7]. PVs are the main source of ectopic activity in AF [32]. Their structure has branching fibres with limited lateral coupling and abrupt changes in their orientation. This provides an adequate anatomic substrate for spontaneous activity and re-entry. In patients with AF, PVs have shown a shorter effective refractory period (ERP), lower voltage, shorter muscle sleeves, slower conduction, and more complex signals [33]. These characteristics make them prone to develop delayed afterdepolarizations (DAD) or early afterdepolarizations (EAD) [11].

A longer APD allows L-type Ca^2+^ channels to recover from inactivation, which causes EAD. This may be due to loss of repolarizing K^+^ current, or a persistent or late Na^+^ current, such as in long-QT syndrome type 3 [34]. DADs are originated mainly from the abnormal release of Ca^2+^ from the SR during diastole, which is then exchanged for extracellular Na^+^ by the Na^+^-Ca^2+^ exchanger type 1 (NCX1) that leads to cell depolarization when it reaches a threshold [22]. Automatism abnormalities may also be a causing or perpetuating factor, since HCN channels that control automatism in the sinoatrial node have shown an increased expression in patients with AF. Nonetheless, their role in this disease is still not fully understood [11].

Regarding the second hallmark, the determinants for conduction are the structural integrity, cell-to-cell coupling, and the rapid phase of Na^+^-current. Alterations in these factors may come from fibrosis and fibroblasts differentiation into myofibroblasts that may promote ectopic activity; mutations of repolarizing K^+^ channels that produce a gain of function, some nucleoporin mutations (*Nup155*); and Na^+^ channel loss of function that can reduce the ERP and lead to a substrate favorable for re-entry [11].

The development of these two hallmarks is independent, and in some patients one of them may be dominant. Most of them respond well to pulmonary vein isolation (PVI), but some may experience recurrence [35]. When AF is established, it generates electrical and structural changes that promote its own maintenance, progression, and stabilization [36,37].

### 2.3. Persistent Atrial Fibrillation

Persistent AF (PeAF) is defined as AF that is continuously sustained beyond 7 days [1]. The progression to this stage has been associated with cardiovascular events, hospitalizations, and death [38]. It has structural and electrophysiological differences by means of which AF can sustain itself and progress [39]. Here, we classify these into four hallmarks: Structural remodeling, electrophysiological remodeling, neurohormonal remodeling, and inflammation.

#### 2.3.1. Structural Remodeling

Fibrosis plays a central role in the pathogenesis of AF. Atrial fibrous-tissue content, and therefore left atrial wall thickness, is increased in patients with PeAF and atrial scarring, and correlates with clinical outcomes [40]. It is classified in reparative and interstitial fibrosis. The first occurs due to the replacement of lost cardiomyocytes, while the second occurs in response to cardiac inflammation or pressure overload, and is sub-classified into reactive fibrosis, which indicates the deposition of extracellular matrix without cell replacement, and infiltrative fibrosis as seen in amyloidosis [16,18].

There are many profibrotic pathways that are upregulated in patients with AF [16]. Increased angiotensin II plays a key role in atrial remodeling by binding to angiotensin receptor 1 (AT1-R) that leads to the activation of the MAPK pathway, which regulates the expression of transforming growth factor β (TGF-β), connective tissue growth factor (CTGF) [41], plasminogen activator inhibitor (PAI-1), and matrix metalloproteinases, that are proinflammatory and profibrotic [16]. TGF-β promotes the synthesis of collagen fibres by cardiac fibroblasts and their differentiation into myofibroblasts. In addition, AT1-R is coupled with G protein that activates phospholipase C to mediate the increase in Ca^2+^ in the cytoplasm, which also promotes fibroblast proliferation and differentiation [42].

Experimental models have demonstrated that suppressing this pathway leads to a reduction in interstitial fibrosis by inhibiting both the mineralocorticoid receptor and the AT1-R [43,44]. This correlates with clinical trials, that have shown the efficacy of the inhibition of this axis for primary prevention of AF, especially in hypertension and heart failure populations, and in preventing AF recurrence after cardioversion and in patients with paroxysmal AF under medical therapy [45,46,47,48], which suggest a potential role for these therapeutic targets in AF secondary prevention. Nonetheless, this has not shown a reduction in mortality [49].

Another key component that suffers from AF remodeling is the sarcomeric cytoskeleton. This organelle supports mechanical contraction, signal transduction, and the transport of ubiquitinated proteins. AF generates mechanical stress and cytoskeletal protein damage that reach a failure in the protein quality control system [12].

Fibrosis leads to AF perpetuation in numerous ways. First, the excess of extracellular matrix creates a physical barrier for conduction, which induces re-entry. Second, the proliferation of fibroblasts and their differentiation into myofibroblasts lead to myofibroblast–cardiomyocyte interaction, that induce re-entry and spontaneous focal activity. Third, slowing conduction and APD reduction induce spontaneous depolarization. These alterations make the ideal substrate for AF.

#### 2.3.2. Electrophysiological Remodeling

The main components of this hallmark include a SR Ca^2+^ overload and Ca^2+^ spontaneous release. Adrenergic stimulation, AT-1R, and oxidative stress cause the activation of the Ca^2+^/calmodulin-dependent protein kinase II (CaMKII) and protein kinase A (PKA), that phosphorylates RyR2 and phospholamban, increasing intracellular Ca^2+^. A reduced expression, hyperphosphorylation of the inhibitory interacting proteins phospholamban or atrial-specific sarcolipin lead to an increase in the activity of SERCA2a [11]. Meanwhile, a high atrial rate causes the accumulation of intracellular Ca^2+^, which activates the Ca^2+^-dependent calcineurin/nuclear factor of activated T cells (NFAT), that leads to a decreased Ca^2+^ current, which shortens the APD. Increased intracellular Ca^2+^ upregulates the expression of K^+^ channels, which also shorten the APD [50]. A reduced APD increases the likelihood of DAD that may lead to spontaneous atrial ectopic activity [31]. The intracellular spontaneous calcium release is favored by RyR2 dysfunction that increases its opening probability. This can be due to loss of RyR2-associated calmodulin, or of juntophilin-2. Its hyperphosphorylation by CaMKII at Ser2814, 2808 or 2030 by PKA promotes RyR2 dysfunction [11].

Uncertainty remains regarding the precise electrophysiological mechanism that sustains AF. Three main mechanisms have been described, and they are not mutually exclusive [51].

Rotors: a re-entry mechanism that consists of a localized circular or spiral wavefront that rotates around an anatomical or functional obstacle, having heterogeneous conduction velocity and an unexcitable center that causes an irregular propagation of electrical activity.Ectopic foci: abnormal regions within the atria that initiate electrical impulses spontaneously or in response to triggers.Multiple wavelets: numerous re-entry circuits in the atria, that can interact with each other, merge, or divide.

#### 2.3.3. Neurohormonal Remodeling

Increased sympathetic nervous system activity has been observed in patients with AF, secondary to an increase in atrial sympathetic nerve densities [14]. β-adrenergic stimulation causes different Ca^2+^ handling effects, including the activation of PKA and CaMKII that can cause EAD, SR Ca^2+^ overload, and Ca^2+^ spontaneous release, with the consecutive increase in spontaneous ectopic activity. Automaticity can be enhanced via α-adrenergic effect, which reduces I_K1_ activity, or for β-adrenergic activation which creates a self-promoting mechanism.

These patients also develop parasympathetic nervous system dysfunction, with a downregulation of M2 receptors, that lead to decreased responsiveness to acetylcholine (ACh) stimulation, and reduced vagal tone due to reduced ACh release from autonomic ganglia which modulate the electrical activity of PV [52].

As previously mentioned, AF leads to an increase in angiotensin II, that activates the MAPK and TGF-β signaling pathways. Another involved hormonal pathway in AF is atrial natriuretic peptide (ANP) dysregulation. ANP bind to natriuretic peptide receptor (NPR) type A, B, and C. NPR-A and -B have been linked with anti-remodeling effects; meanwhile, NPR-C internalizes them for their breakdown in lysosomes [53]. These peptides are involved in inflammation, oxidative stress, endothelial dysfunction, increase in the renin-angiotensin aldosterone system, and hypertension, all of which promote atrial remodeling [54]. Patients with AF have high levels of ANP, which has been associated with adverse outcomes [55].

#### 2.3.4. Inflammation and Oxidative Stress

There is a close relationship between inflammation and the development of AF [15]. Multiple inflammatory mediators like IL-1β, IL-6, IL-8, IL-10, and TNF-α have increased levels in patients with AF [56]. Damaged cells lead to the activation of the NACHT, LRR, and PYD domain containing protein 3 inflammasome (NLRP3), that mediates a proinflammatory response with the release of IL-1β and IL-18. In animal models, NLRP3 has shown Ca^2+^ mishandling, APD abbreviation, and structural remodeling [11], and its genetic suppression prevents the development of AF [57].

Inflammation increases oxidative stress, which can directly affect ion channels and AP. In addition, NADPH oxidase can produce reactive oxygen species (ROS) that lead to myocyte apoptosis, fibrosis, and inflammation. They could also mediate the phosphorylation of RYR2 via CaMKII oxidation, that may provoke Ca^2+^ mishandling [18]. Separately, mitochondria and Ca^2+^ have a close relationship, regulating ATP synthesis and the production of ROS [58].

Another factor that has gained recent interest in AF progression is epicardial adipose tissue (EAT). It covers 80% of the heart’s surface and promotes local inflammation by producing proinflammatory cytokines and adipokines and has been associated with generation of fibrosis [59].

### 2.4. Permanent Atrial Fibrillation

Permanent AF is defined as AF accepted by the physician and the patient when no further rhythm control strategy will be undertaken [1]. This clinical decision may be taken under numerous reasons that go beyond the sole AF progression. There is not a single factor that turns AF from “reversible” to “non-reversible”, but rather the establishment and progression of the aforementioned that evolve until the return to sinus rhythm may not prove beneficial.

Recent trials have shown better outcomes for patients with a restored sinus rhythm compared with only rate control in patients with AF and associated HF or cardiomyopathy [60,61]. Therefore, it is key to aim for an early rhythm control to prevent this cascade of events, and the adverse clinical outcomes for patients.

## 3. Clinical Implications of Atrial Fibrillation

AF is associated with various complications, impaired quality of life, and increased morbidity and mortality. Understanding the clinical implications of AF is crucial for effective management and treatment.

### 3.1. Quality of Life

Impaired quality of life (QoL) has been reported in at least 60% of patients with AF, but only 17% experience disabling symptoms, having a higher impact in women, younger patients, and those with comorbidities [62]. Furthermore, the ORBIT AF cohort study [63], that evaluated patients with the AF-specific quality of life questionnaire (AFECQT) concluded that female sex and a New York Heart Association HF classification III or IV were the factors most strongly associated with impairment in daily activities.

### 3.2. Hospitalizations

Patients with AF have an annual hospitalization rate of 10–40%. It is estimated that 30% of patients are hospitalized once a year, and 10% are hospitalized at least twice a year, representing a two-fold incidence compared to patients in sinus rhythm (37.5% vs. 17.5%) [64]. The most frequent causes of these hospitalizations are stroke (49%), non-cardiac causes (43%), and bleeding (8%) [63].

### 3.3. Increased Risk of Mortality

The risk of mortality is described as 1.5–3.5 times higher compared to patients in sinus rhythm [65]. Additionally, AF patients have other cardiovascular risk factors and comorbidities that independently increase their mortality. In the Framingham Cohort Study [66], AF was associated with an odds ratio (OR) of 1.5 in men and 1.9 in women for mortality, and the risk of AF-related mortality did not significantly vary with age. In the RELY study [67], which randomized 18,113 patients, a total of 1371 deaths occurred within an average follow-up period of 400 days, resulting in an annual mortality rate of 3.84%. The main cause of death was cardiac (37.35%), including both sudden cardiac death (22.25%) and progressive HF (15.1%). Vascular aetiology deaths (10.14%) were mainly reported as due to stroke/peripheral embolism (7%), bleeding (2.84%), and pulmonary embolism (0.29%). Non-cardiovascular causes accounted for 35.81% of deaths and were primarily attributed to cancer (13.93%), respiratory failure (5.76%), infections (4.45%), and indeterminate causes (2.77%).

### 3.4. Stroke and Embolism

Patients with AF have an increased risk of stroke up to 5 times, accounting for 20–30% of strokes, and is suspected of being the cause of most cryptogenic strokes [68]. Ischemic stroke or transient ischemic attack (TIA) represents the initial manifestation of AF in 2–5% of patients [69]. AF-related strokes have a higher risk of disability and mortality when compared to stroke patients in general [70]. Patients with AF also have higher risk of systemic embolism, affecting the lower extremities (60%), the mesenteric region (30%), and upper extremities (10%) [71].

### 3.5. Heart Failure

The incidence of HF has been reported as 33%, 44%, and 56% in patients with paroxysmal, persistent, and permanent AF, respectively [72]. Additionally, both preserved (HFpEF) and reduced (HFrEF) ejection fraction HF are at least twice as common in patients with AF compared to those in sinus rhythm, with HFpEF being more frequently associated [73]. In the Framingham Heart Study, about 30% of HFpEF patients had AF, and it is estimated that 62% of HFpEF patients will develop AF at some point in their lives, which is significantly higher than in HFrEF patient cohorts [74]. In a Swedish HF registry of 41,466 patients, it was concluded that AF was more prevalent with higher ejection fractions. However, patients with AF had a similar increase in the risk of death, HF hospitalization, and stroke compared to patients in sinus rhythm, regardless of ejection fraction [75].

### 3.6. Neuropsychiatric Disorders

Depression has been described in 16–20% of patients, and there is a higher risk of cognitive impairment and vascular dementia (HR 1.4 and 1.6, respectively) independent of prior stroke history [76].

Regarding subclinical AF, the estimated incidence is 30% depending on the screening method used [77]. Its clinical significance is mainly related to three conditions: The risk of cerebral and systemic embolism, arrhythmia burden over a certain period, and the possibility of progression to clinical AF [78].

Given the clinical implications described for AF and considering its increasing prevalence in parallel with population aging, it is currently considered a public health problem with significant associated healthcare costs. Currently, many rhythm control alternatives are available, which will be described in the following sections.

## 4. Classic Therapeutic Approach for Rhythm Control in Atrial Fibrillation

With the passage of time, the importance of rhythm management in the treatment of AF has been emphasized, especially in early stages of the disease [8,9,79,80] and in patients with atrial structural characteristics, that allow for the prolonged maintenance of sinus rhythm over time [10,81,82,83]. Otherwise, in patients in advanced stages and with significant structural remodeling, the focus is on rate control [84,85,86]. Excluding electrical cardioversion and specific surgery procedures, we have two major therapeutic strategies aimed at returning to sinus rhythm, as seen in Figure 2.

### 4.1. Antiarrhythmic Drugs

Developing drugs that will alter the characteristics of the cardiac action potential represents a challenge since most studies concerning efficacy and safety of new pharmacological treatments help to rule out those that cause proarrhythmic events or intracardiac conduction abnormalities [87]. Based on international recommendations for rhythm control in AF with antiarrhythmic medications, three main elements can be derived:

#### 4.1.1. Antiarrhythmic Arsenal

Clinical guidelines recommend the use of a limited number of AAD for the clinical management of patients [1], despite the existence of a wide range of these medications [88], including some that can limit the occurrence of proarrhythmic events, which constitutes the main limitation of the classical drugs used in AF [89]. This is reflected in the use of class I antiarrhythmics such as propafenone and flecainide, and class III antiarrhythmics such as amiodarone, despite the development of several antiarrhythmics over the years [90].

#### 4.1.2. Drug Utilization Algorithm

The indication for the use of these AAD is based on the evaluated cardiac structural characteristics through transthoracic echocardiography [1]. This can be summarized as the utilization of amiodarone in patients with atrial structural abnormalities and reduced ejection fraction (which applies to most of our patients), and propafenone, flecainide, or sotalol for those without significant atrial remodeling and with preserved ejection fraction [88].

#### 4.1.3. Therapeutic Approach

The choice of therapy is related to the potential adverse reactions that these drugs may trigger [1,91], rather than the underlying pathophysiological mechanism underlying the generation or progression of the disease, adapted to each patient and their different stages. This last point reflects a lack of understanding of these mechanisms, which can be explained by various factors resulting in an evident delay in progressing in the understanding of its clinical elements and lacks personalized therapy [12].

### 4.2. Atrial Fibrillation Ablation

Given the difficulty in developing AAD [92], associated with the limited ability to cardiovert and/or maintain sinus rhythm in a significant proportion of patients using classical antiarrhythmics for AF, particularly in the group with permanent AF who experience symptoms that adversely affect their quality of life [1], is that AF ablation was developed. This technique consists primarily of the electrical isolation of the pulmonary vein [93] where linear or circular lesions are created, with or without confirming or mapping trigger foci. It also includes the detection of complex fractionated atrial electrograms (CFAEs) or specific sites where electrical activity in the atria can be mapped, followed by the elimination of these arrhythmogenic substrates through physical methods [94].

This therapeutic strategy has gained popularity due to its positive outcomes in clinical studies for various AF scenarios (as well as for managing other tachyarrhythmias) [95,96,97,98]. Its success rates surpass those of standard approaches such as AAD. However, it is a technique that is challenging to access, expensive [99,100], and requires advanced training [101,102]. Despite these challenges, in experiences at high-volume patient centers, i.e., second-generation cryoballoon ablation significantly reduces the learning curve for junior residents [93,103]. Nevertheless, only a very small percentage of patients have access to this form of disease management. For example, in low-income countries, it is mainly used to treat arrhythmias that have the greatest impact on disability and mortality [104]. This is the reason why AAD will continue to be the most common strategy for controlling sinus rhythm globally, particularly in countries with limited healthcare system access. This highlights the need to refine antiarrhythmic therapeutic targets to make them safer and more effective compared to their predecessors. Additionally, it is crucial to enable more patients to access methods for converting to sinus rhythm, such as ablation, by understanding the mechanisms involved in their utilization.

## 5. Rhythm Control in Atrial Fibrillation: Toward a Translational Approach

### 5.1. Modern Classification of Antiarrhythmics in Atrial Fibrillation

The classical classification of antiarrhythmics proposed by Vaughan-Williams has become limited due to the large number of new molecules that have emerged in recent years. This is why Lei et al. [105] proposed advancing toward a modern classification of AAD that allows for a better classification of the action of each of the new families that have appeared. They suggest a classification from 0 to VII, which also encompasses the modification of these drugs in terms of calcium transient, the proteins involved, and other associated factors, such as intercellular junctions that are still under study. Therefore, a single drug could belong to more than one group of antiarrhythmics.

For this review, we have adapted this new classification to make it specific to the drugs useful in the control of rhythm in AF, as observed in Figure 3. The diagram illustrates the main therapeutic targets for pharmacological cardioversion and maintenance of a rhythm control strategy, which can be modified by known AAD. These drugs are grouped by extending the already known classes or adding new classes of medications. For the rhythm control strategy, two classes of drugs are recognized in the traditional classification, according to international clinical guidelines recommendations [1,88,91,106]. Furthermore, some can be classified within other classes.

#### 5.1.1. Class I

These are AAD whose main mechanism of action is the blockade of the Nav1.5 channel, the primary representative of this family in the human heart [107]. Among the class Ia drugs, we find quinidine and disopyramide, which are less commonly used due to their additional action on potassium channels (Kir, Kv). This group has an incidence of malignant ventricular arrhythmias that can range between 1 and 8% [105]. In the case of class Ic drugs such as flecainide and propafenone, they have sodium channel blocking properties with a different profile than class Ia drugs [108]. Additionally, they antagonize relevant proteins involved in the calcium transient like RyR2, which is why they are also included in class IVb of calcium antagonists [109]. These drugs are ideal for the “pill in pocket” strategy once patients have received therapy with these drugs in the in-hospital setting, demonstrating their safety and efficacy [110]. All class I drugs should not be used in patients with reduced ejection fraction or cardiac structural abnormalities, as it exacerbates the risk of proarrhythmic phenomena [1].

#### 5.1.2. Class III

This group encompasses the largest number of available drugs, as their main function is to reduce the availability of repolarizing currents and, consequently, prolong the APD [108]. The most widely used drug within this group is Amiodarone, given its low cost and high availability. However, it is the oldest and less selective, as it not only blocks voltage-dependent potassium channels but also Na^+^ and Ca^2+^ currents [111]. In addition, it has several complications with chronic use [112]. This makes it significantly proarrhythmic and a preferred choice for patients with reduced ejection fraction and/or structural cardiac abnormalities [1].

Efforts have been made to narrow down the action spectrum of AAD, thus transforming them into increasingly selective targets. This has led to the development of “atrio-specific” drugs [113] such as vernakalant [114], which is a specific inhibitor of the I_Kur_ current (Kv1.5). Although it also reduces the Na_ILate_, another recognized proarrhythmic mechanism [105]. Similarly, a molecule similar to amiodarone, called dronedarone, has been developed with a more restricted spectrum and fewer adverse events [115]. Dronedarone inhibits the I_Ks_ (Kv7.1) and I_Kr_ (Kv11.1) currents.

Finally, there are classic drugs that find their therapeutic niche in patients with preserved cardiac function and no structural abnormalities [88]. These drugs are inhibitors of I_Kr_, such as dofetilide (considered “pure”), ibutilide (associated with Nav1.5 activation), and sotalol (which also blocks Kir2.x, Kv1.4, Kv4.2, and Kv4.3). However, these drugs come with complications, including a high rate of Torsade de pointes, which can exceed 8% [105].

### 5.2. New Antiarrhythmics and Molecular Targets

The limitations of the existing drugs lead to the search for new targets and new approaches for treatment, based on the underlying AF mechanisms. Drugs targeting the ryanodine RyR2 channels, NLRP3, and the development of atrial selective AAD to minimize the occurrence of ventricular proarrhythmia will be discussed [113], as displayed in Figure 4. Other strategies consist of modifying the molecular structure of existing agents (for example, amiodarone). All of this improves its safety and reduces adverse effects [116].

#### 5.2.1. RyR2 Channels

These channels are responsible for spontaneous calcium release from the sarcoplasmic reticulum stores, which activates sodium influx through a Na^+^/Ca^2+^ exchanger (I_NCX_), creating a depolarizing current that generates DAD, leading to ectopic/triggered activity [117]. In human atrial samples, it has been seen that RyR2 dysfunction and increase in calcium release promotes CAMKII, which makes it also an interesting target to reduce ectopic activity (Table 1) [118].

RyR blockers are known as class IVb antiarrhythmics, and we can find some class Ic drugs such as Flecainide and Propafenone [119] and ranolazine, a class Id antiarrhythmic [105] that blocks the late I_Na_ current, affecting action potential recovery, refractoriness, repolarization reserve and QT interval, but it also blocks I_kr_ and it has shown inhibition of atrial selective TASK-1 in vitro studies [120]. As these drugs also inhibit RyR2 channels, they potentially prevent the formation of proarrhythmic Ca^2+^-dependent DADs.

Regarding the synergy of RyR2 blockers with other antiarrhythmics, a study performed in United States, California (completed in 2020) evaluated the effect of ranolazine and dronedarone when given alone and in combination in patients with paroxysmal AF, and showed 0% of mortality and serious adverse effects (NCT01522651).

Another interesting drug is dantrolene, which has shown to be a RyR2 modulator. The effects of dantrolene have been studied in isolated cardiomyocytes from patients with AF, showing that dantrolene was able to reduce SR Ca^2+^ leak and suppress cellular DADs [121]. As for CaMKII blockers that target RyR2 dysfunction, there are in vitro and animal studies for hesperadin and RA608 showing reduction in atrial and ventricular inducibility, but there are no clinical trials yet [113].

**Table 1 ijms-24-12859-t001:** Current and previous studies of RyR2 blockers.

Drug	Study	Primary Outcome Studied	Phase	NCT
Flecainide	To Evaluate the Impact of Oral Flecainide on Quality of Life in Patients with Paroxysmal Atrial Fibrillation	To assess the effect of Flecainide CR on patient-perceived health-related QoL (Quality of Life)	Phase 4	NCT00189319
	Inhalation of Flecainide to Convert Recent Onset SympTomatic Atrial Fibrillation to siNus rhyThm (INSTANT)	Measure efficacy Objective evaluated using ECGs and telemetry to record heart rhythm	Phase 2	NCT03539302
	Predictive Factors to Effectively Terminate Paroxysmal Atrial Fibrillation by Blocking Atrial Selective Ionic Currents (SELECTCARFAP)	Electrocardiographic-based spectral parameters of atrial fibrillatory activity (Dominant frequency) associated with successful or unsuccessful cardioversion in both groups of patients	Phase 4	NCT03005366
	Flecainide Acetate Inhalation Solution for Cardioversion of Recent-Onset, Symptomatic Atrial Fibrillation (RESTORE-1)	Assessment of proportion of patients whose AF converts using continuous ECG monitoring	Phase 3	NCT05039359
	Comparative Study of Flecainide CR and Placebo in the Early Treatment of Atrial Fibrillation	Time to the first relapse after randomization, with or without symptoms documented on ECG, Holter or “Self ECG unit” recording	Phase 4	NCT00408473
	The Use of Flecainide for Treatment of Atrial Fibrillation	Arrythmia free health status	Phase 4	NCT05084495
	Flecainide Versus Amiodarone in the Cardioversion of Paroxysmal Atrial Fibrillation at the Emergency Department, in Patients With Coronary Artery Disease Without Residual Ischemia (FLECA-ED)	The frequency of successful cardioversion to sinus rhythm and The combined frequency of premature ventricular contractions (PVCs), non-sustained ventricular tachycardia (NSVT), sustained ventricular tachycardia (SVT), bradycardia < 50 bpm, and systolic blood pressure < 90 mmHg	Phase 3	NCT05549752
Propafenone	Propafenone in the Treatment of Atrial Fibrillation	Proportion of patients with recurrent AF	Not applicable	NCT03674658
	Antazoline in Comparison to Propafenone in Pharmacological Cardioversion of Atrial Fibrillation (AnProAF)	Conversion of atrial fibrillation to sinus rhythm	Phase 4	NCT05720572
Ranolazine	Ranolazine for the Prevention of Atrial Fibrillation After Electrical Cardioversion (GILEAD)	To determine whether ranolazine is effective in decreasing recurrences of AF in patients with persistent AF successfully treated with electrical cardioversion	Phase 3	NCT01349491
	Study to Evaluate the Effect of Ranolazine and Dronedarone When Given Alone and in Combination in Patients with Paroxysmal Atrial Fibrillation (HARMONY)	Atrial Fibrillation Burden (AFB) at Baseline, Percent Change from Baseline in Atrial Fibrillation Burden (AFB) by Week 12	Phase 2	NCT01522651
	Supression Of Atrial Fibrillation With Ranolazine After Cardiac Surgery	Freedom From Any Episode of Post-Operative Atrial Fibrillation Longer Than 6 h of Duration Occurring During the Study Period	Phase 3	NCT01352416
	Randomized Double Blind Control Trial on Effects of Ranolazine on New-Onset Atrial Fibrillation	Incidence of New-Onset Atrial Fibrillation Rate in Post-Operative Cardiac Surgery Patients	Not applicable	NCT01590979

#### 5.2.2. NLRP3 Inflammasome

NLRP3 inflammasome is a multiprotein complex, responsible for cleavage and release of cytokines IL-1 as well as β and IL-18, which increase atrial dilatation and structural remodeling [122]. A study made in a mice model showed that NLRP3 inflammasome activation was associated with aberrant calcium release, atrial hypertrophy, and shortening of the effective refractory period, and the genetic inhibition of NLRP3 prevented the development of AF [123].

NLRP3 blockers constitute part of the new class VII antiarrhythmics as they target tissue structure remodeling (Table 2) [105]. In this group, we find MCC950, a selective inhibitor and colchicine, a microtubule-disrupting drug used for prophylactic treatment of gout, which disrupts the tubule network that drives the assembly of the NLRP3 inflammasome [57,124]. Other drugs that target inflammation and could reduce atrial remodeling are IL-1 and IL-1ß blockers, such as canakinumab, rilonacept, and anakinra; however, studies should be carried out [113].

#### 5.2.3. Atrial Selective Drugs

These new potential drugs target new channels with predominant expression in atria or distinct electrophysiological properties in atria and ventricles, modifying only atrial function, without other cardiac or systemic actions [125].

Some of the targets that have been studied with this purpose are:Kv1.5 channel: ultra rapid delayed rectifier K current (I_Kur_). These channels are expressed in greater quantities in human atria when compared to the ventricles. They even only contribute to the atrial repolarization since the functional current is detectable only in the atria [126].GIRK1/GIRK4 channels: also known as Kir3.1 and Kir3.4 (I_KAch_). They can be constitutively active or be activated by ACh, increasing K^+^ conductance in the heart. Constitutively active channels develop in the course of AF-related remodeling, predominantly in atrial cardiomyocytes [127]. I_KAch_ hyperpolarizes the membrane and shortens the action potential and effective refractory period, supporting the maintenance of AF and facilitating the occurrence of re-entry [128]. This is why blocking these channels could prolong the APD and ERP, terminating the AF.Ca^2+^ activated Potassium Channels (SK Channels): SK channels (I_SK_) are small-conductance voltage-independent potassium channels that contribute to repolarization [116] and are activated by sub-micromolar concentrations of intracellular free Ca^2+^ ions. Chronic AF is associated with an elevation in the intracellular Ca^2+^ concentration during diastole [120] that may have an impact on SK channel function and its role in promoting AF. The blocking of these channels is expected to prolong the APD and reduce re-entry [117] producing also atrial-selective effects, since SK2 and SK3 channels are predominant in the human atria, and appear to have limited function in ventricles under physiological conditions [118].K_2P_ 3.1 (K_2P_)/TWIK-1, TASK-1, and TASK-3 channels: these are voltage-independent background currents I_TWIK-1_, I_TASK-1_, I_TASK-3_, carried by two pore domain K channels. K_2P_ 1.1, also known as the weak inward rectifying K^+^ channel (TWIK-1) and K_2P_ 3.1, or the TWIK-related acid sensitive K^+^ channel (TASK-1), are the predominant atrial channel subunits, and K_2P_ 3.1 (TASK-1) is preferentially expressed in the human atrium over the ventricle, which suggests atrial selectivity. Chronic AF is associated with an upregulation of the TASK-1 channel expression and function in the atria suggesting that the channels contribute to AF-induced action potential shortening, in order that targeting these could have a potential therapeutic role [129].Atrial-selective Na Channel (I_Na_): The inactivation of the I_Na_ current initiates a refractory period dependent on the duration of action potential, it reduces the membrane excitability, slows down the conduction velocity and the propagation of refractory period, being able to suppress triggered activity and end the re-entry [130]. The possibility to selectively target the atrial tissue is due to the intrinsic functional differences between the atria and ventricular forms of fast sodium channel current I_Na_. For example, the atrial current is inactivated at more negative voltages than the ventricular one, also having a faster onset and a slower recovery from inactivation [131].

For these targets, multiple AADs have been developed (Table 3):Vernakalant: This is an atrial selective class IIIa antiarrhythmic, that acts as a non-selective K^+^ channel blocker [105], targeting the Kv1.5 channel (I_Kur_) and the GIRK channel (I_KAch_), both described above as currents that are present predominantly in atrial tissue, participating in atrial repolarization with little effect on ventricular repolarization. It also has a voltage and rate dependent (fast set, fast offset) I_Na_-blocking effect and blocks the transient outward potassium current I_to_, carried by the Kv4.3 channels [116,132]. The combined effect of the blockade of these channels allows for prolongation of the atrial action potential and refractory period duration, and theoretically minimizes the risk of Torsade de pointes as a result of perturbation of ventricular repolarization [133]. For treating new-onset AF, vernakalant proved its efficacy in three randomized, double-blind trials (ACT I, II, III). In these studies, it led to conversion to sinus rhythm in 51% of the patients (vs. 4% with placebo), and when the efficiency was tested against amiodarone infusion, the results showed a 51.7% cardioversion in vernakalant and only a 5.2% cardioversion with amiodarone was well tolerated, with minimal side effects, with transient hypotension and bradycardia in only 5–10% of patients [134]. According to most recent guidelines, vernakalant is considered one of the most effectively used drugs for cardioversion in patients with AF, even being more efficient and safer than amiodarone as a pharmacologic cardioversion agent as well as flecainide, IV amiodarone, ibutilide, and propafenone [1].Dronedarone: Class IIIa antiarrhythmic [105]. This drug is of particular interest since it is an analog of amiodarone, one of the most used drugs in the management of AF, but it lacks iodine molecules; therefore, it has less pulmonary and thyroid toxicity [135]. In vitro data show that dronedarone inhibits various potassium currents, including: I_kur_, I_KAch_, and K_2P_ channels, all with atrial selective properties. It also targets the transient outward potassium currents I_to_ in human atrial myocytes [136], and concentration-dependent inhibition of sodium currents has been demonstrated utilizing dronedarone in vitro in human atrial myocytes [137]. Addressing the drug security profile, the ATHENA multicenter trial concluded that dronedarone is associated with reduced cardiovascular events in patients with paroxysmal or persistent AF and HF [138]. Additionally, when compared with commonly used drugs, dronedarone was associated with significantly lowered risk of all-cause death than with the use of sotalol, with no differences in AF recurrence observed between the two therapies [139].SK blockers: Class IIIc antiarrhythmics [105]. The current leading SK channel blocker is AP30663l, which has proven to be safe for use in humans in phase 1 studies and has entered phase 2 clinical trials in a pig model AP30663, prolonging the effective refractory period in a dose-dependent manner [140,141].TASK-1 blockers: The respiratory stimulant doxapram acts as a selective TASK-1 blocker, showing antiarrhythmic class III properties. In a pig model, doxapram successfully cardioverted AF [142]. It is currently under study in the doxapram conversion to sinus rhythm (DOCTOS) trial, which will reveal whether doxapram, a potent TASK-1 inhibitor, can be used for acute cardioversion of persistent and paroxysmal AF in patients, potentially leading to a new treatment option for AF [143].Antazoline: First-generation antihistamine, H1 antagonist with a quinidine-like class Ia antiarrhythmic effect [144], targeting I_Na_ channels, which reduces ectopic ventricular/atrial automaticity, accessory pathway conduction and increases refractory period, decreasing re-entry tendency [105,144]. The CANT II study performed in Poland evaluated the efficacy and safety of antazoline, a first-generation antihistamine, for cardioversion of recent onset of AF in the setting of an emergency department. It showed a superior rhythm conversion rate with antazoline when compared to amiodarone and propafenone (78.3% vs. 66.9% and 72.7%) [145]. At the moment, there is a study being conducted also in Warsaw, Poland, comparing antazoline with propafenone in pharmacological cardioversion of AF (NCT05720572).Nifekalant: New class IIIa antiarrhythmic drug approved in Japan for the treatment of ventricular tachyarrhythmias. It is a selective I_kr_ blocker that prolongs effective atrial inactivity. It has no significant effect on myocardial cell conduction velocity or myocardial contractility, which translates in a low incidence of adverse events such as bradycardia and hypotension [146]. When compared with catheter ablation, nifekalant had a better success rate of conversion, with no difference in the incidence of adverse events between the two groups [147].Refralon/Niferidil: New class III antiarrhythmic agent developed in Russia for pharmacological cardioversion. It blocks I_kur_ and prolongs atrial and ventricular action potential and refractory period [148]. In a study of this drug efficacy and safety, there was relief of paroxysmal AF in 95% of the cases, and while in 5% that had a prolongation of the QTc, none of the patients developed Torsade de pointes after administration of the drug [149].

**Table 3 ijms-24-12859-t003:** Current and previous studies of vernakalant, dronedarone, antazoline, nifekalant, and refralon.

Drug	Study	Primary Outcome Studied	Phase	NCT
Vernakalant	RAFF4 Trial: Vernakalant vs. Procainamide for Acute Atrial Fibrillation in the Emergency Department	Conversion to sinus rhythm for a minimum duration of 30 min	Phase 4	NCT04485195
	Predictive factors to effectively terminate paroxysmal atrial fibrillation by blocking atrial selective Ionic Currents (SELECTCARFAP)	Electrocardiographic-based spectral parameters of atrial fibrillatory activity (Dominant frequency) associated with successful or unsuccessful cardioversion in both groups of patients	Phase 4	NCT03005366
	Study of Normal Conditions of Use, Dosing, and Safety of Intravenous (IV) Administration of Vernakalant (MK-6621-049)	Number of Participants Experiencing Significant Hypotension, significant ventricular arrhythmia, atrial flutter and bradycardia	Phase 3	NCT01370629
Dronedarone	Early Dronedarone versus usual care to improve Outcomes in Persons with Newly Diagnosed Atrial Fibrillation (CHANGE-AF)	Cardiovascular Hospitalization or Death	Phase 4	NCT05130268
	Effect of Prolonged Use of Dronedarone on Recurrence in Patients with Non-paroxysmal Atrial Fibrillation After Radiofrequency Ablation	Cumulative non-recurrence rate	Phase 4	NCT05655468
	Study to Evaluate the Effect of Ranolazine and Dronedarone When Given Alone and in Combination in Patients with Paroxysmal Atrial Fibrillation (HARMONY)	Atrial Fibrillation Burden (AFB) at Baseline, Percent Change from Baseline in Atrial Fibrillation Burden (AFB) by Week 12	Phase 2	NCT01522651
	Systematic Review and Meta-Analysis of Multaq^®^ for Safety in Atrial Fibrillation	Number of participants with cardiovascular hospitalization, ventricular proarrhythmia, number of all-cause mortality events, number of participants with atrial fibrillation	Not applicable	NCT05279833
	Early Aggressive Invasive Intervention for Atrial Fibrillation	Time to recurrence of symptomatic or asymptomatic Atrial Fibrillation, Atrial Flutter or Atrial Tachycardia	Not applicable	NCT02825979
	Catheter Ablation vs. Anti-arrhythmic Drug Therapy for Atrial Fibrillation Trial (CABANA)	Number of Participants With Composite of Total Mortality, Disabling Stroke, Serious Bleeding, or Cardiac Arrest in Patients Warranting Therapy for AF	Not applicable	NCT00911508
Antazoline	Antazoline in Comparison to Propafenone in Pharmacological Cardioversion of Atrial Fibrillation (AnProAF)	Conversion of atrial fibrillation to sinus rhythm	Phase 4	NCT05720572
	Antazoline in Rapid Cardioversion of Paroxysmal Atrial Fibrillation (AnPAF)	Conversion of AF to SN confirmed in standard 12-lead ECG during observation period after first iv bolus	Phase 4	NCT01527279
Nifekalant	Comparison of Efficacy and Safety of Different Doses of Nifekalant Instant Cardioversion of Persistent Atrial Fibrillation During Radiofrequency Ablation	Comparison of the successful rates of different doses of nifekalant instant cardioversion of persistent atrial fibrillation after radiofrequency ablation, incidence of adverse effects	Phase 4	NCT04209959
	Nifekalant Versus Amiodarone in New-Onset Atrial Fibrillation After Cardiac Surgery	Rate of cardioversion at 4 h	Phase 3	NCT05169866
Refralon	Efficacy and Safety evaluation of Refralon, concentrate for Solution for intravenous injection in Patients With Paroxysmal and Persistent Atrial Fibrillation and Flutter	Incidence of sinus rhythm restoration	Phase 3	NCT05773170
	Refralon Versus Amiodarone for Cardioversion of Paroxysmal Fibrillation and Atrial Flutter	Restoration of sinus rhythm, sinus rhythm recovery time, recurrent AF after successful cardioversion, ventricular arrhythmogenic effect	Phase 3	NCT05445297
	Refralon in Patients With Recurrence Paroxysmal and Persistent Forms of Atrial Fibrillation Who Underwent Catheter Ablation	Restoration of sinus rhythm, preservation of sinus rhythm, ventricular arrhythmogenic effect, increase QT interval	Phase 3	NCT05456204

### 5.3. Mechanisms Associated with AF Ablation

Ablation for AF was first described in 1998 by Haïssaguerre et al. By performing an intracardiac mapping of isolated ectopic beats, authors were able to highlight several single trigger points located near PVs. Delivering temperature-controlled radio-frequency energy, it was possible to eliminate atrial ectopic rhythm conduction [32].

Different ablation technologies could be exposed, such as radiofrequency, cryoballoon ablation, laser balloon, etc. [150]. It is beyond the scope of this article to describe each of them which are reviewed elsewhere [151]. Briefly, the objective of ablation by different methods is to achieve controlled myocardial damage in specific sites to reduce initiation and maintenance of AF. Here, we discuss some of the proposed mechanisms underlying ablation of AF, as presented in Figure 5.

#### 5.3.1. Electrophysiological Disconnection and Trigger Reduction

Ablation procedure creates lesions or scar tissue. Classical approach consists of a series of point-by-point radiofrequency lesions encircling each or both ipsilateral ostia of the PVs [152]. Achieving electrical disconnection of the PV from the LA has the best clinical results [93,153]; therefore, according to recent guidelines, it is recommended during all AF ablation procedures [93]. The isolation of PV acts as an electrical barrier, blocking the abnormal electrical signals coming from ectopic triggers that can cause AF. Consistent evidence has shown that this technique reduces AF recurrence [154].

As described before, PVs are the main origin of ectopic wavefronts but there are some other sources of abnormal electrical signals [155]. The most common sites are the posterior wall of the LA, the superior vena cava, the crista terminalis, the fossa ovalis, the eustachian ridge, the ligament of Marshall, and adjacent to the AV valve annuli [156]. According to recent guidelines, they should also be ablated when evidenced in electrical mapping [93].

Another approach considers the Bachmann bundle, classically described as an anatomical site of right and left atria electrical connection [157]. Teuwen et al. proposed Bachmann bundle as a potential site with an important role in AF development [158]. Even more, conduction abnormalities at Bachmann’s bundle are related to AF inducibility [159], but there is lack of clinical evidence to recommend it.

Reducing the electrical mass of LA is another proposed mechanism by the effectivity of ablation regarding electrical disconnection [160]. In post-mortem analysis of LA of patients with AF, it revealed an extension of atrial myocardium into 89% of all PVs plus hypertrophy and a higher degree of fibrosis of pulmonary venous myocardium [161]. In the same line, some studies have shown that a large isolation area ablation with verification of the conduction block was proven superior [162] and ostial PVI had better clinical outcomes [163] and it is currently recommended by society guidelines. Despite the clinical importance of this interventional approach, the risk of recurrence is not zero [164]. Ablation seems to be more than trigger prevention only.

#### 5.3.2. Substrate Modification

This approach refers to targeting and modifying the underlying anatomical and electrical abnormalities in the heart that contribute to the initiation and maintenance of AF, which is often related to fibrotic areas. This has been described in the left atrium as a low voltage zone that is often observed in persistent AF [165]. Based on post-mortem findings, fibrosis and fatty infiltration was described in the atrial wall in AF [166], being accurately reflected by electrical mapping as a low voltage zone correlated with magnetic resonance imaging [167,168].

The presence of a low voltage area suggests that the affected region is less electrically conductive and may contribute to the perpetuation or maintenance of the abnormal electrical signals responsible for AF [169]. Therefore, identifying and targeting these low voltage areas in the left atrium has been explored by recent studies as a potential strategy to modify the arrhythmogenic substrate with controversial results. A recent meta-analysis which included retrospective, observational studies plus randomized controlled trials, concluded in favor of this ablation approach in reducing 1 year atrial arrhythmias recurrence [170]. Although, comparing PVI alone versus PVI plus low voltage areas guided substrate ablation in the STABLE-SR II trial concluded no difference in atrial arrhythmia recurrence rate between the two groups [171]. Nevertheless, patient selection could be the cornerstone in substrate modification.

#### 5.3.3. Functional Ablation

This approach focuses on identifying regions within the heart that exhibit abnormal electrical activity by complex mapping strategies. This is often employed when the anatomical source is not entirely clear or as an adjunctive therapy to anatomical ablation. Complex fractionated atrial electrograms (CFAEs) have been classically described as a potential substrate for the genesis and perpetuation of AF. As an electrophysiological finding observed during invasive mapping of the atria in patients with AF, it was focused as an ablation target [172]. This pattern represents an electrical activity characterized by fragmented and disorganized signals recorded from multiple atrial sites and is believed to reflect areas of structural and electrical remodeling within the atria [173]. Although CFAE ablation has a strong rationale, this strategy has been tested in clinical trials with inconclusive results. The START-AF trial demonstrates that ablation of CFAE alone in high-burden AF patients has a low success rate when compared with PV isolation or both procedures combined [174]. Moreover, the START-AF-II trial showed no reduction in the rate of recurrent AF when linear ablation or ablation of complex fractionated electrograms was performed in addition to pulmonary-vein isolation [166].

In addition to CFAEs, dominant frequency (DF) mapping as a functional analysis has been studied. This involves multiple electrodes to record electrical signals in any atrial site and assigning a distinct DF to each of these sites to identify the dominant frequencies or the highest rates of electrical impulses in different regions of the atria, which may represent functional rotors probably responsible for AF maintenance [175].

Sites of high DF should theoretically reflect regions of rotors or drivers of arrhythmia. Atienza and Colls reported successful termination of AF and a significant reduction in dominant frequency, eliminating the left-to-right atrial gradient and in long-term follow-up showed a higher probability of maintaining sinus rhythm [176]. In contrast, when CFAEs and DF have been tested as adjunctive therapy to PVI, there was no improvement in sinus rhythm maintenance [177]. The latest trial RADAF-AF failed to show an additional benefit with high DF ablation [178] and is not recommended by current guidelines.

A different perspective in functional analysis considers localized rotors and focal beat sources. These have been described in animal models as potential sources of fibrillation sites [179]; nevertheless, the pathogenic role in human studies is not fully understood. Although physiologically guided computational mapping revealed sustained electrical rotors and repetitive focal beats during human AF, the CONFIRM trial revealed that ablation is more durable than conventional trigger-based ablation in preventing AF recurrence for 3 years [180].

#### 5.3.4. Autonomic Modulation

Autonomic nervous system modulation has gained special attention. Autonomic modulation has significant effects on cardiac ion channels and subsequently in AF initiation and maintenance [181]. Ablation can target specific autonomic ganglia or perform ganglion plexus ablation to modulate the autonomic influence on the atria [182].

The autonomic system within the heart is organized in clusters of nerve cells located in the epicardial fat pads around the left atrium. This ganglionated plexi (GP) has a significant influence on the autonomic nervous system’s control of the heart rate and rhythm [183]. These clusters contain both sympathetic and parasympathetic nerve fibers and have been related to AF initiation and perpetuation in several studies, showing impact in action potential duration, inward currents and after depolarization within the left atrium [184]. Despite evidence that GP firing is key with or without PV triggers, the AFACT trial showed no improvement in addition to PVI and in fact, showed greater incidence of adverse events [185].

A revolutionary strategy is the catheter-based renal denervation as an adjunctive treatment for AF showing promising results [186]. This technique was first described in patients with resistant hypertension, and the mechanisms behind it are related with the disconnection of renal afferent nerves, which are one of the main regulators of central sympathetic tone [187]. In animal models of AF, renal denervation showed a decrease in renin-angiotensin aldosterone system [188], a decrease in atrial refractory period, and diminished stellate ganglion activity [189]. Prior results have proven the decrease in post-ganglionic sympathetic nerve activity in hypertension patients and decrease in blood pressure [190]. The ERADICATE-AF trial is the largest trial in humans where renal denervation added to catheter ablation was compared with catheter ablation alone. In this multicenter study, an increased freedom from atrial fibrillation at 12 months was reported without increase in adverse events [191].

Despite efforts focusing on a substrate ablation approach, there is still a mechanistic uncertainty of AF. In fact, there are disparities between patients with non-dilated atria with patients with long-standing persistent AF [12]. Finally, focal fibrillation waves occur more frequently at the same site but also occur at more sites [192]. This could explain why the rate of effectiveness is different in early intervention approach.

A better understanding of AF substrate, with better and complex techniques of electrical mapping of the atria, including autonomic regulation, could highlight the relevance of location-guided ablation, which ultimately will traduce better treatment options for patients with paroxysmal or persistent AF.

#### 5.3.5. Techniques for Obtaining Permanent PVI

Several studies and metanalyses have assessed the acute and long-term efficacy of cryoballoon-based PVI. The reported success rate of PV isolation during the procedure is about 98%. Despite this, the long-term effectiveness of the procedure (AF free survival) assessed at 1 year after the ablation is in the range of 70–82%. The AF free survival rate significantly depends on the clinical characteristics of the studied group and the presence of risk factors, especially the type of AF (paroxysmal vs. persistent), LA size, and the presence of heart failure [193]. The indication of catheter ablation using balloon devices has been limited to PVI for paroxysmal AF in Japan, since PVI was thought to be efficient in treating only early-stage (paroxysmal) AF but not efficient enough for treating more advanced (persistent) AF.

The STOP persistent AF trial is a prospective, multicenter, single-arm, FDA-regulated trial designed to evaluate the safety and efficacy of PVI-only cryoballoon ablation for drug-refractory persistent AF in 165 subjects. Among them, 54.8% achieved the primary efficacy endpoint (12 months’ freedom from ≥30 s of atrial tachyarrhythmias), and 86.8% were free from repeat ablation. The primary safety event rate was 0.6%. Significant improvements in QOL and AF-related symptoms were also observed. Based on these results, the indication of cryoballoon ablation was expanded to persistent AF in both USA and Japan in 2020 [194].

On the other hand, the three-dimensional electro-anatomical mapping systems and force-sensing catheters have contributed to reducing radiation exposure during catheter ablation. The use of fluoroscopic imaging integrated with a 3D electro-anatomical mapping system, instead of the conventional electro-anatomical mapping system, during AF ablation was shown to further reduce radiation exposure, without complicating the workflow or compromising acute/mid-term efficacy and safety [195].

### 5.4. Gene Therapy in Atrial Fibrillation

Gene therapy consists of the replacement of a disease-causing gene with a functional one, eliminating a fundamental cause of a particular condition. For this, both the selection criteria and the obstacles to its application in clinical practice must be considered. Along with this, for this therapy to be effective, the desired genetic modification must be expressed in an adequate concentration within the tissue, as well as it must arrive in an appropriate manner [196]. The vehicles used for this process are known as vectors. These can be categorized into two categories, viral, predominantly a retroviral vector, and non-viral, which is the most basic non-viral gene vector and is a naked plasmid DNA directly injected into the myocardium. This type of vector has been used with success in cardiac gene therapy assays and is usually transduced into the cell by the process of lipid-mediated transfection [196]. Unfortunately, to date, there is no vector that is optimal for this purpose; rather, a balance must be found between benefits and harms. In addition, another consideration is to have an appropriate way to reach the target tissue, which is another obstacle for the implementation of this strategy [197]. To reach the atrial and PV tissue, a technique that appears to suitably serve this purpose is gene painting. If further research is pursued, an endocardial gene painting solution may be the most feasible way for electrophysiologists to deliver gene therapy [196]. Despite this, there are different targets that have been studied for the management of this disease, based on their pathophysiological mechanisms. Figure 6 summarizes targets that have been studied.

#### 5.4.1. Targeting Atrial Conduction

Ion channels: Ion channels have been a classic target for the management of AF. In the case of gene therapy, preclinical studies in pigs have shown that the use of an adenovirus with the mutant variant of the CERG-G627S long QT syndrome has been effective in postponing or suppressing the development of persistent AF, through the prolongation of the atrial action potential and its refractory periods [198]. In the same model, blockade of the tandem of P domains in a weak inward rectifying K^+^ channel-related acid-sensitive K^+^ channel-1 (TASK-1), which is an atrial-specific channel for action potential control, from viral vectors carrying anti-TASK-1-siRNA, showed a great decrease in AF burden in the animals studied [129]. The importance of this channel has subsequently been studied as a pharmacological target for AF [199]. Another potassium channel target has been the blockade of the KCNH2 channel through an adenovirus vector carrying KCNH2-G28S, a dominant negative mutation, was shown to prolong the action potential in the porcine model [200], also showing reduction in the same model in the incidence of post-operative AF [201].Ca^2+^ management: Abnormal Ca^2+^ handling is central to the pathogenesis of AF; therefore, efforts to target these mechanisms have been carried out. Wang et al. demonstrated that overexpression of SERCA2a has a suppressive effect on effective refractory period shortening, in addition to AF induced by rapid pacing atrium [202]. The post-transcriptional regulation of RyR2 mediated by the miR-106b-25 cluster could be a potential target for future gene therapy, since its loss would promote paroxysmal AF by the potentially arrhythmogenic Ca^2+^ leakage in the sarcoplasmic reticulum [203].Gap junctions: Modification of the expression of gap junctions is also a critical mechanism of the conduction impairment found in AF. These structures are critical for intracellular conduction by connecting neighboring cells. Downregulation of connexin 43 (Cx43) has been shown to be a contributing factor to AF persistence. In the porcine model, gene transfer of Cx43 has been shown to prevent AF persistence and improve LVEF [204]. Concordant with this finding, Igarashi et al. demonstrated that promoting overexpression of Cx40 and Cx43 connexins preserved ventricular conduction and prevented sustained AF, with no significant differences between the latter [205].

#### 5.4.2. Parasympathetic Stimulus

On the other hand, using plasmid DNA vectors encoding C-terminal fragments of Gαi and Gαo, that inhibit the muscarinic receptor and inhibit interactions between endogenous G proteins, had the effect of disrupting muscarinic signaling in a canine model, thereby decreasing the vagal stimulus-induced AF [206].

#### 5.4.3. Targeting Atrial Remodeling

##### Apoptosis

Strategies to decrease cardiomyocyte cell death, thereby decreasing atrial remodeling, have also been studied, as it is a major concern that atrial dilation and fibrosis could impair the therapeutic effects of this intervention [207]. Caspase 3 knockdown through Ad-siRNA-Cas3 gene transfer in the porcine model via an adenovirus vector, using a hybrid technique between atrial injection followed by epicardial electroporation, increased the expression of the plasmid in the tissue, as well as demonstrated a reduction in apoptosis and prevention of intra-atrial conduction delay [208]. Other targets of this strategy are the decrease in oxidative stress. In cardiomyocytes, overexpression of the micro-RNA miR-206 has been associated with increased production of reactive oxygen species by targeting the enzyme superoxide dismutase 1, in which the damage is associated with remodeling of cardiac autonomic nerves, promoting the genesis of AF [209]. An alternative for the reduction in oxidative stress could be the reinforcement of the antioxidant system through the induction of the Nox family of NADPH oxidases, which has been shown to increase its activity, but not its concentration in atrial tissue, which promotes the production of reactive oxygen species [210].

##### Atrial Fibrosis

Atrial tissue fibrosis is a major contributor to the formation of AF, and TGF-β signaling has been shown to play an important role in the genesis of this tissular alteration. In the canine model, it was proven that injection of a plasmid negative for TGF-β type II receptor substantially decreases atrial fibrosis in the intervening area and improves conduction, which is reflected as the decrease in AF genesis [211]. Another molecular target of interest to combat remodeling is angiotensin-converting enzyme 2, which has been proven that its overexpression in the canine model using an adenoviral vector decreases the levels of collagen I and III in the tissue, in addition to increasing the levels of angiotensin 1–7, which could have cardioprotective effects [212]. This effect is not only associated with slowing down the increase in the expression of MAPK proteins induced by rapid atrial pacing, but also upregulates the protein expression of MKP-1 [213].

## 6. Conclusions

During recent years, the idea of early and sustained restoration of sinus rhythm in patients with atrial fibrillation (AF), has been gaining consolidation. Large clinical studies have been shifting the initial paradigm of indifference between rhythm and rate control as therapeutic strategies in AF, considering the clinical and molecular consequences of maintaining an irregular heartbeat over the years, and how it impacts the disability and mortality of patients. This significant challenge has prompted efforts to refine existing tools, such as pharmacological cardioversion and advances in catheter ablation, in order to develop increasingly personalized and specific therapies while avoiding their inherent adverse effects. To continue progressing toward this objective, it is necessary to understand the foundations supporting each of these approaches, as well as to contemplate the future, including new tools like gene editing, which will likely open up additional therapeutic avenues based on our current understanding of the molecular mechanisms in AF.

## Figures and Tables

**Figure 1 ijms-24-12859-f001:**
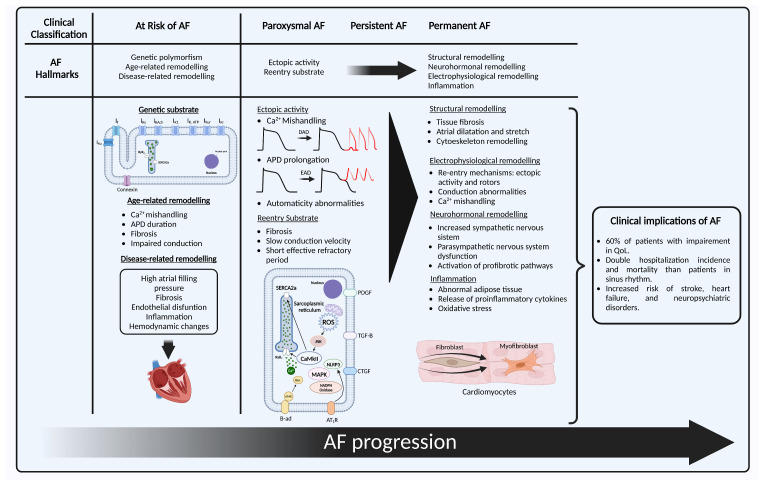
Representation of AF hallmarks with their clinical correlate. First, “at risk of AF” with its three hallmarks: (1) genetic substrate shows a cardiomyocyte with the main structures in which genetic variants for ion channel and non-ion channel genes that have shown association with the development of AF: I_K1_ current (KCNJ2), I_Ks_ (KCNE1, KCNE2, KCNE3, KCNE4, KCNE5, KCND3, KCNQ1), I_KAch_ (KCNJ5), I_Kur_ (KCNA5), I_Kr_ (KCNH2), IKATP (ABCC9), I_f_ (HCN4), INa (SCN1B, SCN2B, SCN3B, SCN5A, SCN10A), connexin 40 (GJA5), connexin 43 (GJA1), and nucleoporin 155 (NUP155); (2) age-related remodeling; (3) disease-related remodeling; next to it the clinical stages of AF, with the initial hallmarks of paroxysmal AF: (1) ectopic activity secondary to DAD, EAD or automaticity abnormalities; (2) re-entry and some of its intracellular mechanisms of perpetuation; and the hallmarks of persistent and permanent AF that include structural, electrophysiological, and neurohormonal remodeling, and inflammation. Created with BioRender.com (accessed on 28 May 2023). AF: atrial fibrillation; DAD: delayed afterdepolarization; EAD: early afterdepolarization; APD: action potential duration; SERCA2a: sarcoplasmic reticulum Ca^2+^-adenosine triphosphate; RyR2: ryanodine receptors; CaMKII: Ca^2+^/calmodulin-dependent protein kinase II; PDGF: platelet-derived growth factor; TGF-β: transforming growth factor β; MAPK: mitogen-activated protein kinase; AMP: adenosine-monophosphate; β-ad: β-adrenergic receptor; AT1-R: angiotensin receptor 1; NLRP3: NACHT, LRR, and PYD domain containing protein 3 inflammasome; ROS: radical oxygen species.

**Figure 2 ijms-24-12859-f002:**
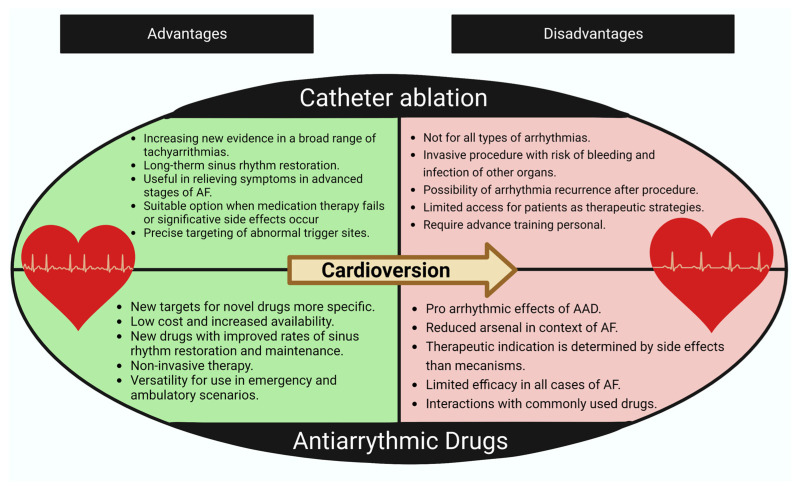
Advantages and disadvantages of the two most common strategies in AF for reversal to sinus rhythm. Created with BioRender.com (accessed on 20 May 2023). AF: arial fibrillation; AAD: antiarrhythmic drugs.

**Figure 3 ijms-24-12859-f003:**
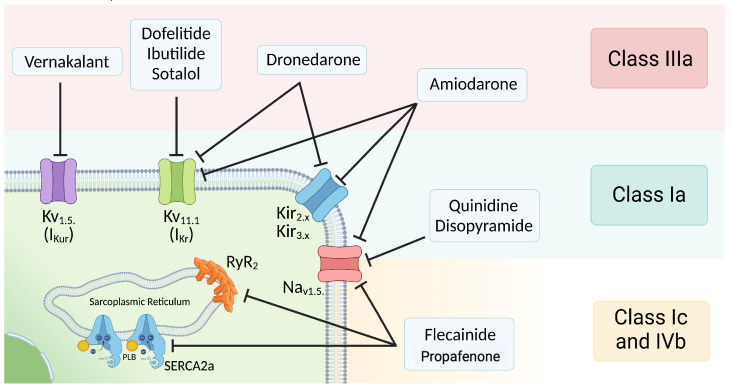
Antiarrhythmics for pharmacological cardioversion in AF. The diagram shows the drugs agreed upon by the largest cardiology scientific societies in their latest clinical guidelines and the main mechanism of action by which they make it possible to modify the action potential and promote the return to sinus rhythm. Although the main channels involved are detailed, these drugs may exert their action with less potency. SERCA2a: sarcoplasmic reticulum Ca^2+^-adenosine triphosphate. Created with BioRender.com (accessed on 20 May 2023).

**Figure 4 ijms-24-12859-f004:**
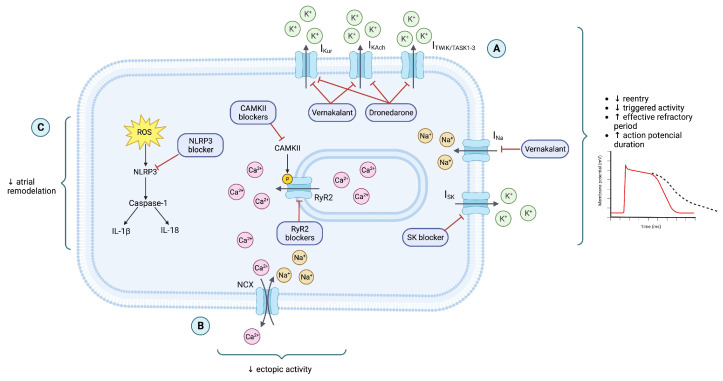
The principal targets that are being studied for AF treatment are atrial selective ionic channels, the RyR2 channel, and the NLRP3 inflammasome. (A) The I_Kur_, I_KAch_, I_TWIK/TASK1-3_, and I_SK_ potassium channels contribute to atrial repolarization and the I_Na_ sodium channel contributes to depolarization. Multichannel blockers such as vernakalant (blocks I_Kur_, I_KAch_, and I_Na_) and dronedarone (blocks I_Kur_, I_KAch_, and I_TWIK/TASK1-3_), TASK-1 and SK blockers, block the ionic flow through these channels, which prolongs the effective refractory period, the action potential duration, and finally reduce triggered activity and re-entry. (B) The RyR2 channel allows for calcium release from the sarcoplasmic reticulum, which leads to sodium influx through the NCX Na^+^/Ca^2+^ exchanger, generating depolarization (specifically delayed after depolarizations). CaMKII phosphorylates RyR2, generating RyR2 dysfunction and an increase in spontaneous calcium release. This can be blocked by RyR2 blockers, RyR2 modulators or CaMKII blockers, all of them leading to a decrease in DADs and ectopic activity. (C) AF is associated with an increase in oxidative stress, ROS production, and NLRP3 inflammasome activation, which is responsible for cleavage via caspase-1 and release of interleukin IL-1β and IL-18, cytokines that are related to atrial structural remodeling. Inflammasome blockers and IL-1β blockers reduce this effect. Created with BioRender.com. AF: atrial fibrillation; DAD: delayed afterdepolarization; IL: interleukin; RyR2: ryanodine receptors; CaMKII: Ca^2+^/calmodulin-dependent protein kinase II; NLRP3: NACHT, LRR, and PYD domain containing protein 3 inflammasome; ROS: radical oxygen species; TASK-1: TWIK-related acid sensitive K^+^ channel; TWIK: weak inward rectifying K^+^ channel.

**Figure 5 ijms-24-12859-f005:**
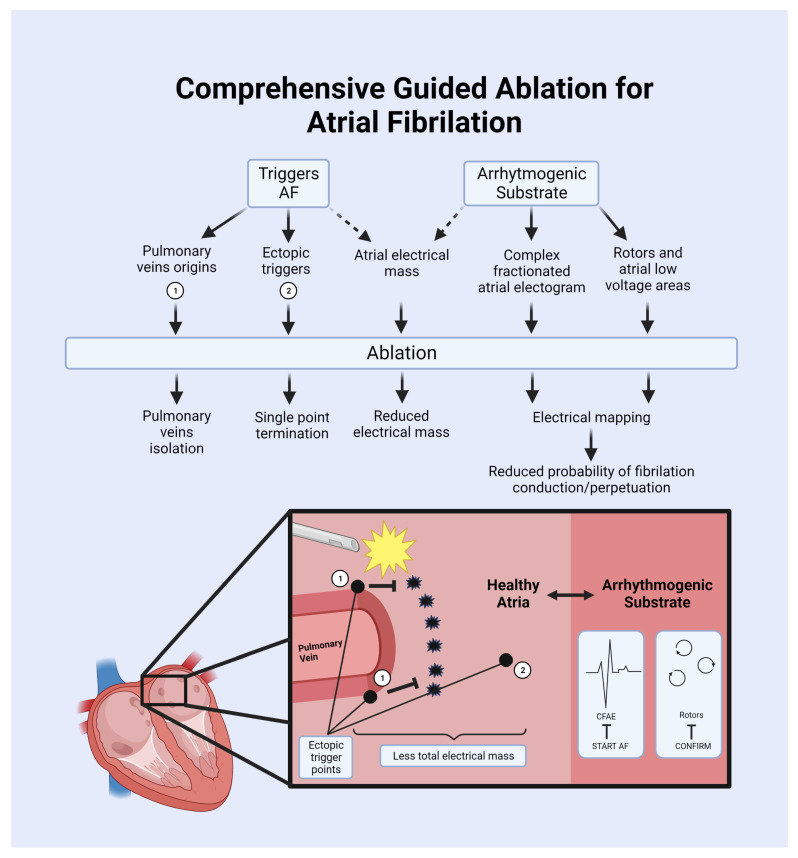
Upper section: approaches to the currently proposed mechanism underlying atrial fibrillation (AF). AF triggers could originate from ectopic regions within the atria or outside, which is the most common location in pulmonary veins ostia. Ablation of triggers point could achieve electrical disconnection of ectopic beats and suppress arrhythmia initiation. Arrhythmogenic substrate could sustain and perpetuate AF. Proposed electrical mapped regions such as complex fractionated atrial electrograms, rotors, and low voltage areas could be ablated to terminate electrical aberrations. Both strategies would reduce the total electrical mass of the atria. Lower section shows an anatomic perspective of ablation mechanism-guided approach to target intrinsic AF pathogenesis. Created with BioRender.com (accessed on 23 May 2023).

**Figure 6 ijms-24-12859-f006:**
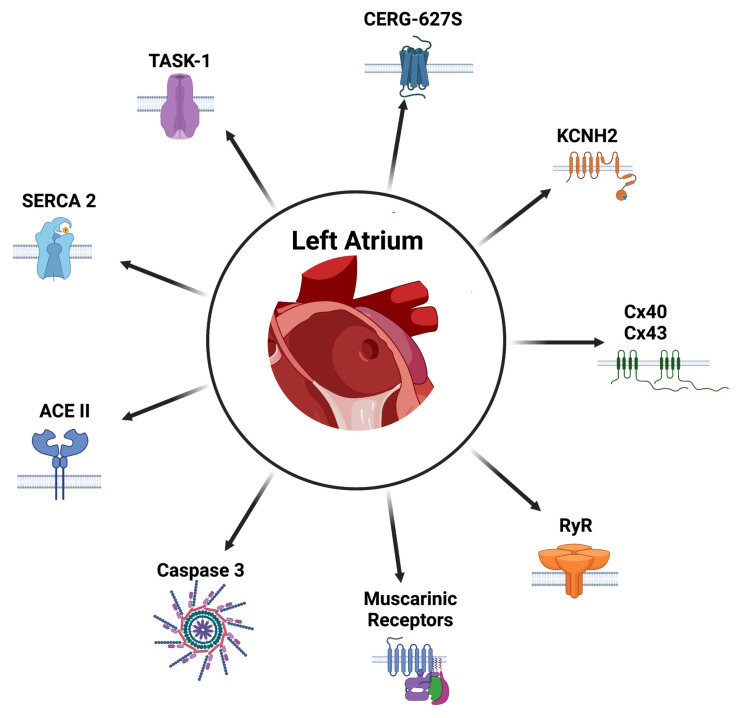
Summary of potential gene therapy targets studied in preclinical studies to date. ACE II: angiotensin-converting enzyme 2; Cx40: connexin 40; Cx43: connexin 43; KCNH2: potassium voltage-gated channel subfamily H member 2; RyR: ryanodine receptor; SERCA2: sarco/endoplasmic reticulum Ca^2+^-ATPase 2; TASK-1: TWIK-related acid-sensitive K^+^ channel 1. Created with BioRender.com.

**Table 2 ijms-24-12859-t002:** Current and previous studies of NL3P blockers.

Drug	Study	Primary Outcome Studied	Phase	NCT
Colchicine	Colchicine and CRP in Atrial Fibrillation and AF Ablation	NCT01755949	Phase 2	NCT01755949
	Use of Colchicine to Decrease Atrial Fibrillation Recurrence After Ablation	Compare atrial fibrillation recurrence and post-ablation quality of life	Phase 3	NCT05459974
	Colchicine in Atrial Fibrillation to Prevent Stroke (CIAFS-1)	Investigate the efficacy of an anti-inflammatory drug, colchicine, at reducing well validated markers of thrombosis (D-dimer) and inflammation (hs-CRP)	Phase 3	NCT02282098
	Colchicine For Prevention of Perioperative Atrial Fibrillation in Patients Undergoing Thoracic Surgery Pilot Study (COP-AF Pilot)	Clinically Significant Atrial Fibrillation	Phase 3	NCT01985425
	Effect of Colchicine on the Incidence of Atrial Fibrillation in Open Heart Surgery Patients (END-AF)	Number of participants with Atrial fibrillation and adverse effects of colchicine	Phase 3	NCT03021343

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
