# Peer review of "Tic-Tac: A Translational Approach in Mechanisms Associated with Irregular Heartbeat and Sinus Rhythm Restoration in Atrial Fibrillation Patients"

_ijms, 2023, doi:10.3390/ijms241612859_

Round 1
Reviewer 1 Report
This review article by Parra-Lucares et al. focused on the mechanisms associated with irregular heartbeat and sinus rhythm restoration in patients with atrial fibrillation (AF). Authors reviewed many preceding literatures and explained each mechanism to develop AF, especially focusing on genetic factors. As authors mentioned, is has been recognized that early restoration of sinus rhythm is important to maintain atrial function and to improve outcomes, since catheter ablation has been widespread. Therefore, the concept of this review article to overview about it is valuable and all contents seem agreeable and educational. Since this manuscript seems written well, I do not have major concern to be resolved.
Author Response
Dear Sir or Madam,
We express our heartfelt gratitude for your thoughtful and positive evaluation of our review article. Your kind words regarding the quality of our writing and the educational nature of the contents are greatly appreciated. It is our aim to present a comprehensive and well-structured review, and your positive feedback serves as encouragement to continue our dedication to providing valuable contributions to the field.
We sincerely thank you for your time and effort in reviewing our manuscript. Your favorable assessment and lack of major concerns provide us with valuable reassurance that our work has met the standards of the journal. Your constructive comments have undoubtedly enriched the quality of our article.
Yours faithfully,
Alfredo Parra-Lucares, MD PhD
Division of Critical Care Medicine
Hospital Clinico Universidad de Chile
Luis Toro, MD PhD FACP
Division of Nephrology
Hospital Clinico Universidad de Chile

Reviewer 2 Report
I went carefully through the review written by Parra-Lucares and co-workers. According to the manuscript’s title the authors probably aimed to explain the mechanisms related with the irregularity of heart rate and restoration of sinus rhythm in AF patients. It seems to me that this title is completely unclear to readers and it must be changed. Moreover, as far as I am concerned, the manuscript does not have any clearly defined goal.
In the abstract the authors define the aim of study as “outlining the primary mechanisms involved in the maintenance of AF and their modification when restoring sinus rhythm through the main therapeutic strategies currently in different phases of clinical investigation and to addresses future therapeutic approaches, including gene therapy”. However in the main article it is defined as to present “the current basic-clinical overview of rhythm restoration at the molecular level in all the current and future therapeutic strategies based on preclinical studies, as well as those still under advanced research, both in pharmacological approaches and those associated with electrophysiological ablation. Furthermore to provide a future perspective based on gene therapy in AF, emphasizing the importance of early reversal from irregular and chaotic rhythm to sinus rhythm in AF patients”
Furthermore over 50% of the manuscript neither explains primary (molecular) mechanisms involved in the maintenance of AF nor shows how they change with restoration of sinus rhythm. It seems to present some clinical and basic science knowledge about AF with emphasis on some molecular phenomena, often clearly unproven. Unfortunately this manuscript is badly designed and written. Finally this paper requires significant editing, as it is not written in sound English and cannot be accepted in its current form.
This paper requires significant editing, as it is not written in sound English and cannot be accepted in its current form.
Author Response
Dear Sir or Madam,
We appreciate your review of our manuscript, but we must express our concern about the tone and lack of specificity in your comments.
While we welcome constructive criticism, the characterization of our work as "badly designed and written" is not accompanied by specific examples or actionable feedback. Providing concrete points of improvement would have been more valuable and allowed us to address your concerns effectively.
Additionally, your skepticism regarding the validity of certain molecular phenomena in the manuscript appears unsubstantiated. We assure you that our research is based on a rigorous review of existing literature and scientific evidence. If you have specific references or evidence to support your claims, we would be grateful if you could share them with us, enabling us to conduct a thorough review and make necessary revisions.
We also encourage you to consider the broader context of the manuscript's reviews. While you may have reservations, it is worth noting that two other esteemed reviewers highly praised the work and acknowledged its value to the field.
Given this balanced perspective, we cannot disregard their positive evaluations and accept your negative comments without proper justification.
Regards,
Alfredo Parra-Lucares, MD PhD
Division of Critical Care Medicine
Hospital Clinico Universidad de Chile
Luis Toro, MD PhD FACP
Division of Nephrology
Hospital Clinico Universidad de Chile

Reviewer 3 Report
Alfredo Parra-Lucares reviewed ″Tic-Tac: A translational approach in mechanisms associated with irregular heartbeat and sinus rhythm restoration in Atrial Fibrillation patients”. This is a review that covers a wide range of issues, from the mechanism of atrial fibrillation development, new drug, gene therapy and catheter ablation. Especially the molecular biology approach is well organized and the manuscript is well written. However, there are some defects must be addressed.
Concerning #1
In the section of 2. The Hallmarks of Atrial Fibrillation, the authors described in detail the mechanism from the onset of atrial fibrillation to the persistence by atrial remodeling. They summarized a lot of information in a concise and well-written. However, according to a systematic review by MP Schneider et al, RAS inhibitors such as ACEi and ARB were shown to have a primary prevention effect on AF in hypertension and heart failure populations, however the effect on secondary prevention was not proven. This suggests that RAS inhibitors may contribute to suppressing the occurrence of triggers rather than atrial remodeling in clinical practice, this is an unsolved paradox. The discussion on this issue is insufficient in this section.
Concerning #2
In the section 3.6. Neuropsychiatric Disorders, there is a description of the ASSERT study on CIEDs patients. Is this appropriate? It is probably a mistake in the section division. In addition, the ASSERT study is a special cohort targeting CIEDs patients and is not suitable for mentioning atrial fibrillation totally.
Concering #3
Regarding the section 4.2 Electrophysiological Ablation, the progress of AF ablation since 2015 has been remarkable, and it is not possible to judge the ablation issue base on the paper published in 2015. Currently, the procedure of PVI using CRYO balloon has been simplified, so it can be performed by junior residents. Advanced training is not always necessary. Please revise this section.
Concering #4
Regarding the section 5.3.2. Substrate modification, it goes without saying that the effectiveness of trigger ablation by PVI, however the method of modifying the arrhythmogenic substrate of AF has not been established yet. The sentence about CFAE ablation, which is almost denied now, is too long. Rather, you should refer to the low voltage zone often observed in the left atrium of persistent atrial fibrillation and the ablation for it.
Author Response
Dear Sir or Madam,
Thank you for the revision of our article and the comments on our manuscript. We believe that your review helped us to improve the quality of the manuscript and clarify issues that were not appropriately detailed. Accordingly, we are sending a revised version of our manuscript, including the corrections you required. As you will notice, we agree with your comments and concerns, and we are sending you a point-by-point response to your comments.
Thank you for your time and allowing us to resubmit a new version of our manuscript. We look forward to hearing from you and responding to any further questions and comments you may make
Yours faithfully,
Alfredo Parra-Lucares, MD
Division of Critical Care Medicine
Hospital Clinico Universidad de Chile
Luis Toro, MD PhD FACP
Division of Nephrology
Hospital Clinico Universidad de Chile
Comments and Suggestions for Authors:
- In the section of 2. The Hallmarks of Atrial Fibrillation, the authors described in detail the mechanism from the onset of atrial fibrillation to the persistence by atrial remodeling. They summarized a lot of information in a concise and well-written. However, according to a systematic review by MP Schneider et al, RAS inhibitors such as ACEi and ARB were shown to have a primary prevention effect on AF in hypertension and heart failure populations, however the effect on secondary prevention was not proven. This suggests that RAS inhibitors may contribute to suppressing the occurrence of triggers rather than atrial remodeling in clinical practice, this is an unsolved paradox. The discussion on this issue is insufficient in this section.
Thank you for your thoughtful review. We appreciate your valuable insights and have carefully considered your comments to enhance the clarity and strength of our discussion on the involvement of the renin-angiotensin-aldosterone system (RAAS) in atrial remodeling.
As you rightly pointed out, the exact mechanisms by which the RAAS is implicated in atrial remodeling remain paradoxical. However, we acknowledge that there is a wealth of experimental evidence supporting the role of this pathway in atrial fibrillation (AF) and fibrosis, both in primary and secondary prevention.
In response to your suggestions, we have made significant revisions to Section 2.3.1. We have incorporated the results of Schneider et al. to bolster our discussion on this topic. Their findings align with the theme of our research and provide valuable additional insights into the involvement of RAAS in atrial remodeling.
Furthermore, to provide a more comprehensive analysis, we have included supplementary information that delves deeper into the molecular and cellular aspects of the RAAS pathway, as well as its interactions with other signaling cascades implicated in atrial remodeling. This additional information aims to furnish a more well-rounded understanding of the complex processes underlying AF and fibrosis.
We believe these modifications significantly strengthen the rationale for the role of the RAAS in atrial remodeling, and we hope that these enhancements address your concerns and contribute to the overall quality of the manuscript.
- In the section 3.6. Neuropsychiatric Disorders, there is a description of the ASSERT study on CIEDs patients. Is this appropriate? It is probably a mistake in the section division. In addition, the ASSERT study is a special cohort targeting CIEDs patients and is not suitable for mentioning atrial fibrillation totally.
Thank you for your insightful review. We appreciate your keen attention to detail, and we apologize for the mistake in the section division and reference inclusion regarding the ASSERT study.
In response to your comment, we have promptly rectified the issue. The paragraph and reference concerning the ASSERT study have been removed from the manuscript as they are not relevant to the specific population we are addressing in our research. This revision ensures that the focus of our study remains clear and coherent, without any irrelevant information that may cause confusion or misinterpretation.
We thank you for bringing this matter to our attention, and we believe that the manuscript is now more accurate and precise without the inclusion of the extraneous material. We are committed to delivering a high-quality and rigorously researched article, and your valuable feedback has undoubtedly contributed to the improvement of our work.
- Regarding the section 4.2 Electrophysiological Ablation, the progress of AF ablation since 2015 has been remarkable, and it is not possible to judge the ablation issue base on the paper published in 2015. Currently, the procedure of PVI using CRYO balloon has been simplified, so it can be performed by junior residents. Advanced training is not always necessary. Please revise this section.
Thank you for your valuable review. We appreciate your feedback regarding the section on Electrophysiological Ablation (Section 4.2). We acknowledge the significant progress in AF ablation since 2015. We agree that it would not be appropriate to base our judgment solely on a paper from that year, and we have revised the section to reflect the latest advancements in AF ablation techniques beyond 2015.
We also agree with your observations on the simplification of the Pulmonary Vein Isolation (PVI) procedure using CRYO balloon, making it more accessible to junior residents without advanced training. However, we wish to clarify that while certain electrophysiology procedures have been simplified, we respectfully disagree with the notion that mere simplification alone is sufficient to expand their coverage in the global healthcare setting. Access to electrophysiology procedures remains limited in different regions worldwide due to healthcare system coverage and the lack of training programs. Our revised section now emphasizes the importance of addressing these disparities to facilitate the broader implementation of AF ablation techniques.
We sincerely appreciate your careful evaluation of our manuscript, which has allowed us to refine our research and discussion. We hope that the updated section adequately addresses your concerns and contributes to the overall quality of the paper.
- Regarding the section 5.3.2. Substrate modification, it goes without saying that the effectiveness of trigger ablation by PVI, however the method of modifying the arrhythmogenic substrate of AF has not been established yet. The sentence about CFAE ablation, which is almost denied now, is too long. Rather, you should refer to the low voltage zone often observed in the left atrium of persistent atrial fibrillation and the ablation for it.
Thank you for your valuable feedback on our manuscript. We sincerely appreciate your insights. In response to your suggestions, we have made the following revisions: 1) We have included a comprehensive discussion on the low voltage zones and their ablation as a treatment approach for persistent atrial fibrillation; 2) The sentence about PVI has been deleted as it is not directly related to the current focus; 3) The paragraph regarding CFAE has been appropriately shortened for conciseness. We believe these modifications significantly enhance the manuscript's quality and address your concerns effectively. Once again, we are grateful for your thoughtful review, and we hope that these revisions meet your expectations and contribute to the overall strength of the paper. Your continued guidance and support are highly appreciated as we strive to improve and refine our research.

Round 2
Reviewer 2 Report
Dear authors,
The manuscript has improved a lot after revision. I am happy to see the clearly defined goal of review. However, there are still caveats that must be corrected. It seems to me that the fragments of manuscript concerning invasive AF treatment need intensive editing. I strongly encourage to read guidelines written by Calkins H et al. (2017 HRS/EHRA/ECAS/APHRS/SOLAECE expert consensus statement on catheter and surgical ablation of atrial fibrillation. Heart Rhythm)
1. Electrophysiological Ablation – it is better to trim this to AF ablation as in the majority of cases AF ablation is an anatomical not electrophysiological procedure
2. “A therapeutic strategy has been developed to reduce the burden of arrhythmogenic mechanisms through the local application of cold or heat” – actually lesion creation can be achieved with many energy sources such as RF energy, cryoenergy, laser, ultrasound or even radiation. Moreover “cold or heat” sounds unprofessional. Furthermore those energy may be delivered with linear, balloon or other catheters.
3. “This technique involves mapping the sites where AF trigger foci are located for initial identification, followed by the elimination of these arrhythmogenic substrates through physical methods” - in the majority of cases AF ablation does not involve the mapping of trigger foci but it is the totally empirical isolation of pulmonary veins. However extra PV triggers can be mapped.
4. “However, it is a technique that is challenging to access, expensive, and requires advanced training. Despite these challenges, in experiences at high-volume patient centers, second-generation cryoballoon ablation significantly reduces the learning curve for junior residents.”- provide the success rate of AF ablation that is totally the same for the different strategies. Cryoballoon ablation is not superior to other techniques.
5. It must be highlighted that PVI is the cornerstone of all ablation strategies in AF (IA) and other strategies have low level of recommendation (IIB)
6. Techniques for obtaining permanent PVI could be achieved with balloon technologies under X-ray or under 3D electroanatomical guidance.
7. Substrate modification – it refers to targeting detected areas of fibrosis, based either on voltage mapping or on MRI. CFAEs or rotors are functional phenomena so it is functional ablation.
8. Additional (to PVI) ablation strategies has been developed to improve the outcomes of ablation such as targeting detected areas of fibrosis, mapping and ablation of rotational activity, localization and ablation of left atrial ganglionated plexi, dominant frequency mapping, renal denervation and finally nonablative strategies. All of them should be included in the manuscript.
Author Response
See attached document.

Reviewer 3 Report
The authors have addressed essentially all my previous comments, and their revisions have substantially improved the manuscript. I have no further comments.
Author Response
Reviewer 3 – Round 2
Dear Sir or Madam,
We want to express our gratitude for the time and effort the reviewer dedicated to reviewing our manuscript. Their feedback has been immensely helpful in refining our research. We are pleased to know that our revisions have met their expectations.
Yours faithfully,
Alfredo Parra-Lucares, MD PhD
Division of Critical Care Medicine
Hospital Clinico Universidad de Chile
Luis Toro, MD PhD FACP
Division of Nephrology
Hospital Clinico Universidad de Chile
